# Antioxidative Potential and Ameliorative Effects of Rice Bran Fermented with *Lactobacillus* against High-Fat Diet-Induced Oxidative Stress in Mice

**DOI:** 10.3390/antiox13060639

**Published:** 2024-05-24

**Authors:** Tingting Yin, Yidan Chen, Wenzhao Li, Tingting Tang, Tong Li, Binbin Xie, Dong Xiao, Hailun He

**Affiliations:** 1School of Life Sciences, Central South University, Changsha 410083, China; tingyin96@163.com (T.Y.); chenyidan293@163.com (Y.C.); mx20010403@163.com (W.L.); 212511065@csu.edu.cn (T.T.); 8202201019@csu.edu.cn (T.L.); 2State Key Laboratory of Microbial Technology, Shandong University, Qingdao 266237, China; xbb@sdu.edu.cn; 3State Key Laboratory of Coal Resources and Safe Mining, China University of Mining and Technology, Xuzhou 221116, China; xd@cumt.edu.cn

**Keywords:** *Lactobacillus fermentum*, rice bran, antioxidant, transcriptomic analysis

## Abstract

Rice bran is an important byproduct of the rice polishing process, rich in nutrients, but it is underutilized and often used as feed or discarded, resulting in a huge amount of waste. In this study, rice bran was fermented by *Lactobacillus fermentum* MF423 to obtain a product with high antioxidant activity. First, a reliable and efficient method for assessing the antioxidant capacity of the fermentation products was established using high-performance liquid chromatography (HPLC), which ensured the consistency of the batch fermentation. The fermented rice bran product (FLRB) exhibited significant antioxidant activity in cells, *C. elegans*, and hyperlipidemic mice. Transcriptome analysis of mouse livers showed that the expression of *plin5* was upregulated in diabetic mice administered FLRB, thereby preventing the excessive production of free fatty acids (FFAs) and the subsequent generation of large amounts of reactive oxygen species (ROS). These studies lay the foundation for the application of rice bran fermentation products.

## 1. Introduction

Oxidative stress is intricately associated with human health, primarily stemming from the excessive generation of free radicals within the body, including hydroxyl radicals, peroxyl radicals, superoxide anions, and others, subsequently leading to oxidative damage to the organism [1]. For example, disorders of lipid metabolism in type II diabetics lead to the overproduction of free fatty acids (FFAs). The metabolism of these excess FFAs in the liver generates large amounts of ROS, which can lead to liver injury [2]. Therefore, reducing the production of ROS and protecting the liver from oxidative damage is crucial for patients with T2DM [3]. Growing research has demonstrated that antioxidants, especially natural antioxidants, could effectively protect the liver by reducing the concentration of ROS [4].

Rice bran, which is replete with proteins, dietary fiber, and a spectrum of nutrients, is an essential raw material for the development of functional foods. Despite its nutritional wealth, its utilization rate in processing is strikingly low, with a significant portion being used as poultry feed or discarded, thereby undervaluing its potential [5]. As a copious byproduct of rice milling, rice bran is often underutilized, and systematic research on its value-added applications is lacking. *Lactic acid bacteria* (LAB), renowned for their probiotic benefits, are increasingly being recognized for their ability to improve food safety and promote health and are commonly used in food processing. Fermentation with LAB promotes the release of active compounds in food resources, which has clearly been shown to improve the bioavailability of rice bran [6,7]. Multiple studies have shown that the fermentation products of LAB can improve the gut microbiota, reduce cholesterol and blood sugar levels, and alleviate hypertension [8]. In addition, fermentation by LAB can also significantly enhance the antioxidant capacity of fermented products. Our previous study revealed that the antioxidant, hypolipidemic, and hypoglycaemic activities of rice bran products were significantly elevated after fermentation with LAB [9]. Consequently, LAB fermentation is a promising strategy for enhancing the utilization and value of rice bran. In this study, *Lactobacillus fermentum* MF423 was utilized to ferment rice bran to obtain fermentation products with high antioxidant activity, and a rapid and effective fingerprinting assay method was established using high-performance liquid chromatography to ensure the stability of the fermentation products. Then, the antioxidant effects of MF423 fermentation products were analyzed using cellular, wild-type *Caenorhabditis elegans* (*C. elegans*) N2 and high-fat mouse models, and the potential value of rice bran fermentation products in ameliorating diabetes-induced liver injury was explored, which lays the foundation for the application of *Lactobacillus* rice bran fermentation products.

## 2. Materials and Methods

### 2.1. Materials

Rice bran was purchased from a produce market in Changsha, Hunan, China. The *Lactobacillus plantarum* 793 strain is a laboratory-preserved strain, while *Lactobacillus casei* MF439 and *Lactobacillus fermentum* MF423 were isolated from rice flour fermentation broth. Human umbilical vein endothelial cells (HUVECs) were generously provided by Prof. Xiang Rong’s research group at the School of Life Sciences, Central South University, and human dermal fibroblasts (HDFs) and human immortalized keratinocyte (HaCaT) cells were obtained from Professor Su Juan’s research group at Xiangya Medical School, Central South University. *C. elegans* and *Escherichia coli* OP50 (*E. coli* OP50) were obtained from the Caenorhaditis Genetics Center (CGC). Beef extract, tryptone, and yeast extract were purchased from Oxoid (Basingstoke, UK). RPMI 1640 and DMEM, as well as fetal bovine serum, were purchased from Biand (Viva Cell, Shanghai, China). All the chemical reagents used in this study were of analytical grade.

### 2.2. Preparation of Fermentation Medium and Fermentation Broth

The *Lactobacillus* strains were cultured in MRS media. Three 2% (*v*/*v*) *Lactobacillus* solutions (OD = 1.0) were inoculated separately into fermentation media. The rice bran fermentation medium was composed of 5 g of rice bran, 0.1 g of Na_2_HPO_4_, 0.03 g of KH_2_PO_4_, 0.1 g of CaCl_2_, 0.1 g of Na_2_CO_3_, and 50 mL of distilled water [10]. The three *Lactobacillus* strains were incubated at 37 °C, and the fermentation broth was collected by centrifugation (12,000 rpm, 30 min, 4 °C) at various time points. Then, the supernatant was stored at −80 °C.

### 2.3. Comparison of the Antioxidant Activity of the Fermentation Products and Optimum Fermentation Time for Different Strains

The hydroxyl radical-scavenging activity was determined according to the method of Wang [11]. The groups with rice bran products fermented by different *Lactobacillus* strains were used as the test group (As). The group without any antioxidants was used as the injured control (Ai), while the mixture without H_2_O_2_ was used as a blank (Ab). The hydroxyl radical-scavenging activity (HRSA) was quantified using the following equation: HRSA (%) = ((As − Ai))/((Ab − Ai)) × 100%. The strains and fermentation times with higher hydroxyl radical-scavenging activity were selected for the follow-up study.

### 2.4. Composition Analysis of MF423 Fermentation Product from Rice Bran

Ultra-high-performance liquid chromatography–tandem mass spectrometry (UHPLC-MS/MS) was employed to analyze the compositional changes before and after fermentation. The supernatant was collected after high-speed centrifugation of the crude fermentation broth and subsequently filtered through a 0.22 μm membrane before analysis. Chromatographic conditions included an Agilent ZORBAX Eclipse Plus C18 column (3.0 × 100 mm, 1.8 μm), with the mobile phase composed of organic phase acetonitrile (A) and aqueous phase (B) containing 0.05 mmol/L formic acid (mass spectrometry grade). The flow rate was maintained at 0.4 mL/min, and the injection volume was 2 μL. Mass spectrometry detection conditions utilized a positive ionization mode, with electrospray ionization (ESI) as the ionization method. Prior to sample analysis, Agilent Standard Tune Mix ESI-L Low Concentration Tuning Mix (G1969-85000) was employed for accurate mass calibration. The primary mass spectrum scan detection range was *m*/*z* 100–1700. Nitrogen gas was used as the drying gas, with a temperature of 325 °C and a flow rate of 6.8 L/min. The sheath gas temperature was set to 350 °C, capillary voltage was 4.0 kV, and fragment voltage was set to 150 V. HPLC gradient elution program is available in Appendix A.

### 2.5. The Fermentation Products Were Detected by HPLC

The fermentation broth was first subjected to centrifugation at 12,000 rpm, and the supernatant was further filtered through a 0.45 μm filter membrane, and 0.1% trifluoroacetic acid was added to the test sample (50 µL). The determination was performed on a C18 column with mobile phase consisting of water–acetonitrile–methanol solution at the column temperature of 30 °C and the flow rate of 1 mL/min. The UV absorption detection wavelengths were set at 260, 280, and 310 nm [11].

### 2.6. Cell Cytotoxicity and Intracellular ROS Assessments

The cytotoxicity of fermentation products on cells was evaluated by a 3-(4,5-dimethylthiazol-2-yl)-2,5-diphenyl-tetrazolium bromide (MTT) assay. The methodology used for the MTT assay was consistent with the procedure outlined in our previous study (Wu et al., 2018) [12]. Different cellular models of oxidative stress were established by stimulating HUVECs with 35 mM glucose, HaCaT cells with 1.5 mM H_2_O_2_, and HDF cells with UVA, followed by fluorescence imaging of the cells using a Cytation 5 (Biotek, Winooski, VT, USA). The protective effect of FLRB on oxidative damage in the three cell lines was then assessed. A reactive oxygen species assay kit (Beyotime, Shanghai, China) was used to investigate intracellular ROS formation. Additionally, the intracellular radical-scavenging activity of the fermentation products was determined according to methods outlined in our previous study [12].

### 2.7. The Protective Effect of FLRB on Oxidative Stress in C. elegans

#### 2.7.1. *C. elegans* Treatment Protocol

Fresh Nematode Growth Medium (NGM) plates were prepared and inoculated with standard *E. coli* OP50 and *E. coli* OP50 containing varying concentrations of FLRB (0.5 and 1 mg/mL). These plates were incubated at 37 °C for 48 h, creating blank food plates and sample food plates for *C. elegans* cultivation. *C. elegans* synchronized to the L4 stage were individually transferred to both the blank and sample plates. Subsequently, on a daily basis, the *C. elegans* were relocated to new plates with corresponding food sources to prevent interference from newly hatched larvae. After three days of treatment, the *C. elegans* were ready for subsequent experimentation.

#### 2.7.2. Toxic Effects of Fermentation Products on *C. elegans*

Age-synchronized *C. elegans* (L4 stage) worms were individually inoculated onto blank plates and sample plates (each containing 100 worms). The plates were then cultured in a biochemical incubator at 20 °C, and the initiation of the longevity assay was marked as Day 0. *C. elegans* worms were observed daily, and their mortality status was recorded. Every 3 days, *C. elegans* were transferred to corresponding new NGM plates until all nematodes had perished. Survival rates of the *C. elegans* worms were calculated, and survival time was plotted on the *x*-axis against the nematode survival rate on the *y*-axis. This allowed for the observation of the impact of different sample concentrations on nematode longevity.

#### 2.7.3. Influence of FLRB on the ROS Levels and Activities of Antioxidant Enzymes in *C. elegans*

After 3 days of drug treatment, the *C. elegans* worms were transferred to M9 buffer solution containing 100 μM DCFH-DA probe and incubated at 20 °C for 30 min in the dark. Excess probe was removed by washing with M9 buffer solution, and 20 μL of a 25 mM NaNO_3_ solution containing anesthetized nematodes was added. Then, 20 μL of the suspension was placed on a glass slide, followed by the addition of 5 μL of an antifade fluorescence quencher. The ROS fluorescence intensity in the *C. elegans* worms was observed using an inverted fluorescence microscope [13]. Then, the fluorescence intensity was measured at 15 min intervals for 6 h with an excitation wavelength of 485 nm and an emission wavelength of 530 nm at 25 °C. The catalase (CAT), superoxide dismutase (SOD), and glutathione peroxidase (GSH-Px) activities and malondialdehyde (MDA) levels in the *C. elegans* worms were measured by means of kit assays according to the manufacturer’s instructions.

#### 2.7.4. Influence of FLRB on the Antioxidative Stress Capacity of *C. elegans*

After 3 days of drug treatment, *C. elegans* were transferred onto NGM plates supplemented with 10 mM H_2_O_2_ or 10 mM methyl viologen, with 50 worms per group. The number of surviving nematodes was counted at 30 min intervals (on H_2_O_2_ plates) and at 12 h intervals (on methyl viologen plates) until all the nematodes had died. *C. elegans* survival curves were plotted using survival time as the *x*-axis and survival rate as the *y*-axis, enabling a comprehensive assessment of the impact of the fermentation products on the antioxidant stress capacity of the *C. elegans* worms.

### 2.8. Animals and Experimental Design

A total of 60 SPF-grade C57BL/6J male mice were randomly divided into two groups based on body weight: the control group (10 mice) and the model group (50 mice). The control group was fed a low-fat diet, while the model group was fed a high-fat diet. After 8 weeks of diet treatment, the mice were fasted for 12 h and then received a single intraperitoneal injection of a low dose of streptozotocin (STZ) at 35 mg/kg. Blood glucose levels were measured three days later, and mice with blood glucose levels greater than 6.0 mmol/L were considered insulin-resistant model mice. Healthy mice fed a low-fat diet served as the normal control group and were continuously fed a low-fat diet. The model mice were randomly divided into different groups based on body weight: model group (HFD), pioglitazone group (PGLT), rice bran nonfermented extract group (RB), high-dose rice bran fermented extract group (HFLRB), and low-dose rice bran fermented extract group (LFLRB). The mice in these groups were continuously fed a high-fat diet. The treatment for each group started in the 8th week of the experiment, with daily gavage administration for a duration of 8 weeks. At the end of the experiment, all the mice were sacrificed. The liver tissues were then dissected from the mice. A portion of the liver tissue was fixed in a tissue fixation solution, while another portion was preserved in RNA protection solution and stored at −80 °C for subsequent analysis.

### 2.9. Transcriptomic Analysis

#### RNA Isolation, Library Construction, and Sequencing

Three liver tissue samples from the control group (NC) were compared with an additional three samples from the HFLRB group. After the mice were euthanized, total RNA was extracted and purified from the liver tissues, followed by quantification and quality assessment. Afterwards, six RNA samples were packed in dry ice and delivered to Majorbio Biopharm Technology Co. (Changsha, China) for further library construction and sequencing on an Illumina HiSeq 2500 platform. Gene quantification was conducted employing the RSEM software tool (http://deweylab.biostat.wisc.edu/rsem/, accessed on 9 April 2021), followed by data normalization utilizing the fragments per kilobase per million mapped reads (FPKM) and transcripts per million reads (TPM) methods. Thereafter, the identification of differentially expressed genes across the various samples was performed using the edge R package (http://www.r-project.org/, accessed on 9 April 2021). This analytical process adheres to established procedures in transcriptomics, encompassing quantification, normalization, and differential gene expression analysis. Next, differential gene expression analysis between samples was conducted using DESeq2 software (http://bioconductor.org/packages/stats/bioc/DESeq2/, accessed on 9 April 2021). Gene Ontology (GO, https://github.com/tanghaibao/GOatools, accessed on 30 March 2022) and Kyoto Encyclopedia of Genes and Genomes (KEGG, http://www.genome.jp/kegg/pathway.html, accessed on 30 March 2022) enrichment analyses of the differentially expressed genes were performed with the cluster profile R package.

### 2.10. Real-Time Fluorescence Quantitative PCR

Total RNA from the liver tissues of the normal control group, model group, and high-dose fermentation product-treated mice was extracted using the EZB Total RNA Extraction Kit (EZBioscience, Roseville, MN, USA). Following the instructions of the reverse transcription kit (Vazyme, Nanjing, China), RNA was reverse transcribed into cDNA. Gene expression analysis was performed using the Bio-Rad CFX96 Touch Real-Time PCR detection system for quantitative reverse transcription PCR (qRT-PCR). Primers for RT-PCR were designed online using NCBI-Gene, and their specific sequences are listed in Appendix A.

### 2.11. Statistical Analysis

The experiments were performed in triplicate, and the results are expressed as the mean ± standard deviation (SD) unless otherwise specified. The statistical analysis of the experimental data was conducted using Origin 9.1 software, while one-way analysis of variance (ANOVA) was performed using SPSS 19.0 software. The significance level was set at *p* < 0.05, which was considered to indicate statistical significance.

## 3. Results and Discussion

### 3.1. Comparison of the Antioxidant Activity of Fermentation Products and the Optimum Fermentation Time for Different Strains

A comparison of the hydroxyl radical-scavenging abilities of fermentation products from three strains of *Lactobacillus* at different time points showed that the fermentation product of *Lactobacillus fermentum* MF423 reached a significantly greater antioxidant activity at 24 h (55.18%) (Figure 1). Both the fermentation time and antioxidant activity of *Lactobacillus fermentum* MF423 were notably superior to those of the other two strains of *Lactobacillus*. Therefore, all subsequent experiments utilized fermentation products (FLRB) obtained by fermenting rice bran with *Lactobacillus fermentum* MF423 for 24 h.

### 3.2. HPLC Detection of FLRB

To assess the antioxidant activity of the fermentation products more comprehensively, we prepared multiple fermentation batches of rice bran using MF423. Additionally, we performed multiple HPLC analyses on the same samples to validate the stability of the HPLC detection method. Through HPLC analysis of multiple batches of the MF423 fermentation product, it was determined that the HPLC method was characterized by a high degree of stability when used to monitor the fermentation product (Appendix A). Notably, a conspicuous increase in the intensity of the absorption peak with a retention time of 15.9 min was detected at both 220 nm and 280 nm postfermentation compared to prefermentation. Conversely, at all three tested wavelengths, the retention time of the component at 16.7 min significantly decreased, indicating the consumption of certain substances during MF423 fermentation. This decrease was attributed to the consumption of the substance during the MF423 fermentation process. By employing gel filtration chromatography, we successfully isolated and purified the components of the MF423 fermentation product, specifically selecting those with high antioxidant activity. These components with high antioxidant activity were subsequently subjected to HPLC analysis. A distinctive absorption peak with a retention time of 15.9/16.0 min emerged as a hallmark of the product. This peak exhibited robust hydroxyl radical-scavenging capacity. Therefore, the antioxidative potency of the fermentation product could be directly gauged by monitoring the intensity of the absorption peak at 15.9/16.0 min.

### 3.3. Component Analysis of FLRB

To further elucidate the changes that occurred in the products of *Lactobacillus* rice bran before and after fermentation, we analyzed the compositional profiles of the products using UHPLC-MS to examine the changes in substance composition. The postfermentation peaks (Appendix A) showed significant changes in substance composition compared with the prefermentation peaks (Appendix A) in the time range of 4–12 min. There was a significant increase in the composition of the substance in the postfermentation product at 4.395 min, in addition to new substances detected in the fermentation product at 6.248 min, 8.249 min, 10.319 min, and 11.565 min. In the unfermented rice bran, there was a significant decrease in the fractions at 14.426 min, 14.598 min, and 19.600 min, indicating degradation or transformation of the fractions. Based on the molecular formula analysis, the main compounds generated during fermentation are outlined in Table 1. Yi et al. confirmed an increase in phenolic compounds in rice bran fermented with *Lactobacillus plantarum*, which exhibited increased DPPH radical-scavenging ability and ferric-reducing antioxidant power (FRAP) [14]. In our current study, fermenting rice bran with *Lactobacillus fermentum* MF423 resulted in a notable increase in cinnamic acid, which is a typical phenolic compound. It is postulated that this phenolic compound contributes to the antioxidant activity observed in fermented rice bran products [15]. Phenolic compounds, known for their elevated redox potential, exhibit hydrogen donating and metal ion chelating properties, rendering them valuable as antioxidants [16]. Additionally, the fermentation process produces short peptides rich in antioxidative amino acids, such as Pro, Phe, His, and Tyr [17]. The mass spectrometry results showed that the rice bran fermentation products were rich in antioxidant components, which may help to enhance their antioxidant properties.

### 3.4. Protective Effect of FLRB against Cellular Oxidative Stress

Endothelial oxidative injury is closely associated with the occurrence of many diseases, such as arteriosclerosis and diabetes. Therefore, enhancing the resistance of endothelial cells (such as HUVECs) to oxidative damage is critical for mitigating and treating various vascular dysfunction-related conditions. HaCaT cells are immortalized keratinocytes, which constitute the outermost layer of human skin, known as the stratum corneum. Oxidative damage to keratinocytes directly impacts human health [18]. Human dermal fibroblasts (HDFs) reside in the dermal layer of the skin and are the primary cells involved in the synthesis and secretion of collagen protein [19]. UVA can penetrate through the atmosphere and reach the dermal layer of the skin, directly damaging fibroblasts and inducing cellular aging and oxidative stress, which may even result in various skin disorders [20]. To further investigate the antioxidant activity of the MF423 fermentation products, we established oxidative stress models using HUVECs, HDFs, and HaCaTs to determine the antioxidant function of MF423. The cytotoxic effects of different concentrations of fermented products on HUVECs, HaCaTs, and HDFs were investigated through an MTT assay. The results revealed that the fermentation products did not show toxicity to the three different cell types in the concentration range of 25–200 μg/mL (Appendix A). To investigate the effects of intracellular radical scavenging, ROS accumulation in cells was determined by DCFH-DA. HUVECs, HaCaTs, and HDFs were stimulated with high glucose, H_2_O_2_, and UVA, respectively, after which elevated intracellular ROS fluorescence was observed, indicating that the oxidative stress models had been successfully established. When the cells were treated with the unfermented product (negative control), no ROS scavenging effect was observed. In contrast, when the concentration of FLRB reached 50 μg/mL, it effectively reduced the ROS levels in HUVECs and HaCaT s to normal levels (Figure 2a–d). As previously mentioned, 1.25 g/L rice-fermented *Lactobacillus plantarum* significantly reduced the ROS level in HDFs after UVA irradiation to normal levels [21]. However, in our study, 25 μg/mL of the fermentation product was able to significantly remove ROS from HDFs (Figure 2e,f). These results suggested that FLRB possesses notable antioxidant properties and can attenuate oxidative stress in various cell types.

To gain further insight into the antioxidant mechanism of FLRB at the cellular level, we assessed the activity of antioxidant enzymes in cells treated with FLRB. The results indicated that the application of fermentation products to HUVECs led to an increase in both SOD and GSH-Px enzyme activity while also counteracting the reduction in CAT enzyme activity caused by high glucose (Figure 3a–c). Furthermore, treatment of HaCaT cells with the fermented product significantly enhanced the activities of SOD and CAT enzymes while mitigating the decrease in GSH-Px enzyme activity caused by H_2_O_2_ stimulation (Figure 3d–f). Similarly, treatment of HDFs with the fermented product alleviated the decrease in SOD enzyme activity induced by UVA exposure and markedly increased the activities of CAT and GSH-Px enzymes in the cells (Figure 3g–i). In summary, MF423 fermentation enhanced the antioxidant capacity of cells by increasing the activity of relevant antioxidant enzymes within the cells.

### 3.5. The Protective Effects of the MF423 Fermentation Product on Oxidative Stress in C. elegans

In the commonly utilized *C. elegans* model for assessing antioxidant activity, the accumulation of excessive ROS leads to oxidative stress and accelerates the aging process. Therefore, investigating the impact of FLRB on the lifespan of *C. elegans* provided evidence of the antioxidant efficacy of FLRB on the nematode lifespan. After the nematodes were cultured with different concentrations of the fermentation product, it was observed that *C. elegans* treated with a 1.0 mg/mL concentration of the fermentation product exhibited an extended lifespan (Figure 4a). To further verify whether the fermentation product exhibits antioxidative effects in vivo and to evaluate its antioxidative efficacy, we utilized H_2_O_2_ and methyl viologen to stimulate *C. elegans* to establish an oxidative stress model. This model was utilized to further confirm the antioxidative properties of the fermentation product within the organism. As shown in Figure 4, when the concentration of the fermentation products increased to 1.0 mg/mL, it significantly inhibited the oxidative stress induced by H_2_O_2_ in *C. elegans*, thus slowing their mortality rate (*p* < 0.01) (Figure 4b). The fermentation products exhibited certain protective effects against oxidative stress induced by methyl viologen (*p* < 0.05) (Figure 4c). Compared with those in the control group, treatment with 0.5 mg/mL FLRB resulted in a significant reduction in ROS levels in *C. elegans*, with the average fluorescence intensity decreasing by 69.32%. In addition, the fluorescence accumulation intensity of *C. elegans* was measured using a microplate reader, and the results are shown in Figure 4d. The fluorescence accumulation in *C. elegans* was consistent with the average fluorescence intensity obtained from the statistical analysis of the ROS images. The results collectively demonstrated the potent ROS scavenging ability of FLRB in *C. elegans*, effectively reducing the elevated ROS levels within the nematode. This scavenging effect may be a contributing factor to the observed decrease in the mortality rate of *C. elegans*. MDA, a product of lipid peroxidation, can interfere with the mitochondrial respiratory chain complex and key enzyme activity and, therefore, serves as an indicator of antioxidant effects. The results depicted in Figure 4g–j indicate that when the concentration of the fermentation product was 1.0 mg/mL, the content of MDA in *C. elegans* was reduced to 33.27% of that in the control group, while the activity of SOD increased by 1.92 times and that of GSH-Px increased by 1.43 times; the CAT activity remained unchanged. In summary, MF423 fermentation products exhibited several beneficial effects, including extending the lifespan of *C. elegans*, suppressing oxidative stress induced by H_2_O_2_ and methyl viologen, reducing malondialdehyde levels, inhibiting lipid peroxidation, and upregulating the activities of SOD and GSH-Px. These activities collectively enhanced the antioxidant capacity of *C. elegans*.

### 3.6. Comparison of Mouse Liver Transcriptomes between the Control Group and the FLRB-Group

Diabetes, which is typically triggered by a combination of genetic and environmental factors, manifests in two main types, with Type II being the more prevalent type. Due to lipid metabolism disruption leading to an excess of free fatty acids (FFAs) in type II diabetes patients, the surplus FFAs undergo metabolism within the liver, generating a considerable amount of reactive oxygen species (ROS). This process disrupts the balance of the liver’s oxidative environment, ultimately resulting in hepatic damage. Consequently, safeguarding the liver from oxidative injury has become a crucial therapeutic approach [3]. The emergence of insulin resistance, for instance, precipitates a notable surge in intracellular ROS and lipid peroxidation end-product MDA levels within the hepatic milieu [22]. Existing investigations have firmly established the presence of a myriad of endogenous antioxidant enzymes, including superoxide dismutase (SOD), total antioxidant capacity (T-AOC), and GSH-Px, which effectively mitigate oxidative stress by scavenging reactive oxygen species [23]. Our previous research demonstrated that FLRB could effectively ameliorate disturbances in glucose and lipid metabolism in type 2 diabetic mice [9]. Eight weeks of FLRB treatment significantly reduced the levels of blood glucose and lipids and elevated antioxidant activity in type 2 diabetes mellitus (T2DM) mice. H&E staining revealed alleviation of overt lesions in the livers of FLRB-treated mice. Additionally, FLRB also demonstrated the capability to enhance hepatic activities of GSH-Px and SOD, thereby augmenting the overall antioxidative capacity of the liver. To further investigate the correlation between the antioxidant capacity of the fermentation product and its effects on lowering blood sugar and lipids, we conducted a transcriptomic analysis of the liver in type 2 diabetic mice treated with the fermented product. A total of 446 genes were identified as differentially expressed genes (DEGs) in the FLRB group compared to the control group (*p* < 0.01), 180 of which were upregulated and 266 of which were downregulated (Appendix A). GO analysis suggested that DEGs in the FLRB group were significantly enriched in several GO BP terms related to lipid metabolism, such as acylglycerol metabolic process, neutral lipid metabolic process, icosane metabolic process, and response to insulin (Table 2). KEGG analysis revealed enrichment in seven major pathways associated with lipid metabolism. Notably, the fermented product showed the greatest enrichment in genes related to the peroxisome proliferator-activated receptor (PPAR) signaling pathway. This pathway holds significant prominence in lipid metabolism research, serving as a pivotal signaling cascade within lipid metabolic processes (Figure 5). Interestingly, among the genes enriched in this pathway, the gene encoding perilipin5 (*plin5*), a protein involved in lipid droplet coating, stands out. The *plin5* can effectively prevent the accumulation of ROS due to excess FFAs by inhibiting the hydrolysis of triglycerides (TGs), thereby reducing the damage to the liver caused by oxidative damage [24]. The notable upregulation of *plin5* suggested that the fermentation product might play a role in reducing oxidative damage in the liver by affecting the transcription of this gene [25].

Real-time quantitative PCR was used to assess the changes in the expression of the differentially expressed gene *plin5* and antioxidant-related enes. The result was consistent with the transcriptomic analysis showing that the fermentation product significantly upregulated the expression of *plin5*, nuclear factor erythroid 2 related factor-2 (*nrf2*), haem oxygenase 1 (*ho1*), glutamate cysteine ligase catalytic subunit (*gclc*), and superoxide dismutase 2 (*sod2*) in the liver (Figure 6a–f). The upregulation of *plin5* promotes the storage of liver fat in the form of triglycerides, thereby preventing the excessive production of free fatty acids (FFAs) and subsequent overoxidation metabolism, leading to the generation of ROS. Additionally, studies have shown that the high expression of *plin5* in pancreatic β-cells can further activate the Nrf2/ARE antioxidant signaling pathway. The Nrf2/ARE signaling pathway serves as a crucial molecular pathway during oxidative stress responses in organisms (Figure 7). This activation potentially promotes the expression of downstream target genes, resulting in the heightened expression of relevant antioxidant enzymes, such as SOD, as well as an increase in the synthesis of GSH, thereby facilitating their antioxidative functions within the organism [23].

## 4. Conclusions

This study investigated *Lactobacillus fermentum* MF423, a strain that ferments rice bran into products with potent antioxidant capabilities. An HPLC fingerprinting technique was developed to monitor the antioxidant activity of these fermentation products, ensuring reliable and direct assessment. The antioxidant efficacy of the fermentation product was confirmed through cellular tests involving various human cell types and through experiments with nematodes, where it was shown to boost antioxidant enzyme activity, suggesting potential anti-aging benefits. Our previous research indicated that FLRB can effectively ameliorate dysregulated glucose and lipid metabolism in type 2 diabetic mice, which is closely associated with an increase in the overall antioxidant capacity of the liver [10]. To determine the potential mechanisms by which the antioxidant properties of the fermentation product could lead to hypoglycemic and lipid-lowering effects, we performed a transcriptomic analysis on the livers of type 2 diabetic mice treated with the fermentation product. The findings revealed that the fermentation product, by upregulating the expression of the *plin5* gene, facilitated the storage of triglycerides in the liver to prevent the overoxidation of free fatty acids and the consequent generation of ROS. Additionally, the fermentation product activated the PI3K/AKT signaling pathway through *plin5*, which subsequently activated the Nrf2/ARE signaling pathway, enhancing the expression of related antioxidant enzymes downstream and thus protecting the livers of high-fat diet-fed mice from oxidative damage. In conclusion, this study developed an HPLC fingerprinting technique that provides a solid basis for the industrial-scale production of rice bran fermented by strain MF423. However, further investigations into the optimal conditions for industrial fermentation processes are needed. Moreover, additional research is essential to clarify the precise mechanisms through which the critical antioxidant molecules in the fermentation product can upregulate the *plin5* gene and regulate the NRF2/ARE signaling pathway.

## Figures and Tables

**Figure 1 antioxidants-13-00639-f001:**
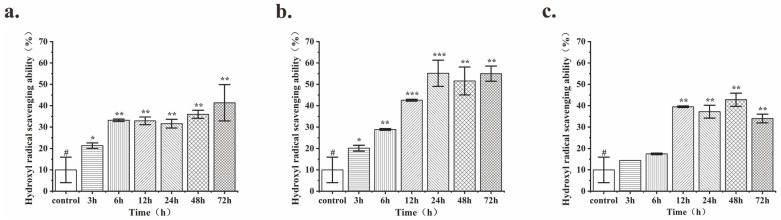
Hydroxyl radical scavenging activity was assessed in the products of fermented rice bran by different *Lactobacillus* strains at various fermentation times. *Lactobacillus plantarum* (**a**); *Lactobacillus fermentum* (**b**); *Lactobacillus casei* (**c**); “^#^” represents the comparison with the unfermented group, “*”, “**”, and “***” represents *p* < 0.05, *p* < 0.01, and *p* < 0.001.

**Figure 2 antioxidants-13-00639-f002:**
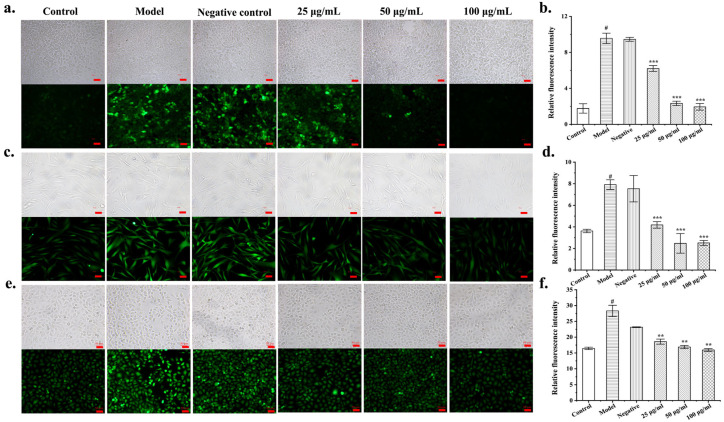
Protective effects of MF423 fermented rice bran products against cellular oxidative stress. Intracellular ROS on HUVECs (**a**), HaCaT (**c**), HDFs (**e**) cells indicated as green DCFH-DA fluorescence. Images were taken using fluorescence microscope (magnification, 10×). Statistical analysis was performed to quantify relative fluorescence density of HUVEC (**b**), HaCaT (**d**), HDF (**f**) cells. “^#^” means *p* < 0.01 compared with control; “**”, and “***” representing *p* < 0.01, and *p* < 0.001. compared with model.

**Figure 3 antioxidants-13-00639-f003:**
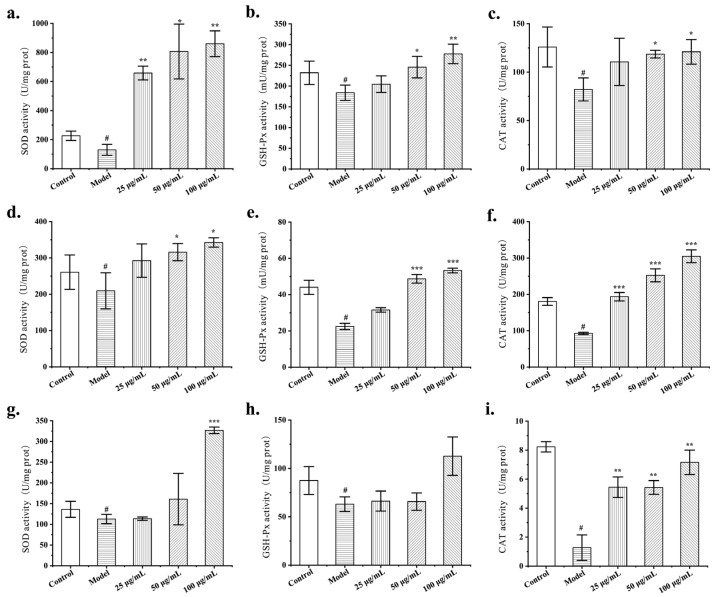
The effect of FLRB on SOD, GSH-Px, and CAT enzyme activity in cells. HUVECs (**a**–**c**); HDFs (**d**–**f**); HaCaT (**g**–**i**). “^#^” means *p* < 0.01 compared with control; “*”, “**”, and “***” representing *p* < 0.05, *p* < 0.01, and *p* < 0.001 compared with model.

**Figure 4 antioxidants-13-00639-f004:**
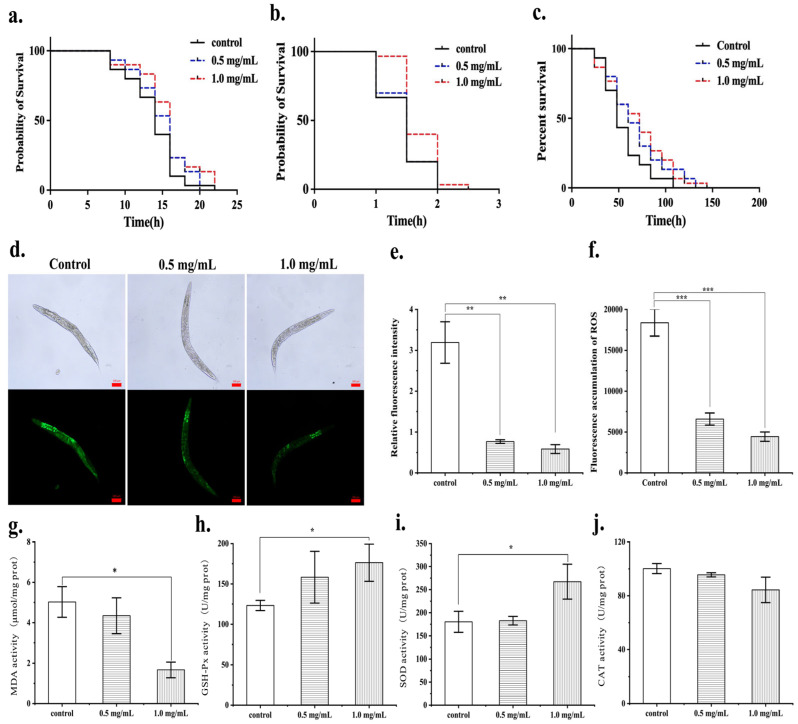
The protective effects of the fermentation product of MF423 on oxidative stress in *C. elegans*. (**a**) The effect of fermented rice bran product (FLRB) on the lifespan of *C. elegans* (n = 50 worms). (**b**,**c**) Effects of FLRB on the antioxidant stress response in *C. elegans* (n = 50 worms); H_2_O_2_-induced oxidative stress model (**b**). (**c**) Methyl viologen-induced oxidative stress model. (**d**–**f**) The scavenging effect of FLRB on reactive oxygen species (ROS) in *C. elegans* (n = 3 groups). Fluorescent images of ROS in worms (100× magnification) (**d**). Comparison of average fluorescence intensity (**e**). Comparison of fluorescence accumulation (**f**). (**g**–**j**) The malondialdehyde (MDA) content and antioxidant enzyme activity in *C. elegans*. MDA (**g**); glutathione peroxidase (GSH-Px) enzyme activity (**h**); superoxide dismutase (SOD) enzyme activity (**i**); catalase (CAT) enzyme activity (**j**). “*”, “**”, and “***” representing *p* < 0.05, *p* < 0.01, and *p* < 0.001.

**Figure 5 antioxidants-13-00639-f005:**
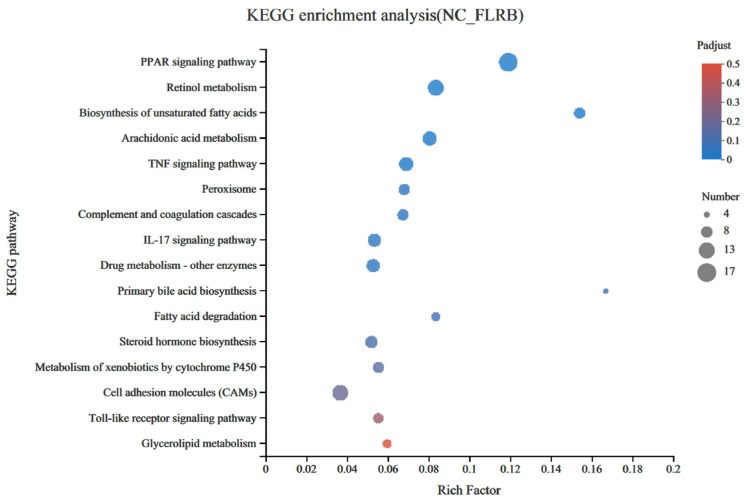
Kyoto Encyclopedia of Genes and Genomes (KEGG) signaling pathway enrichment analysis.

**Figure 6 antioxidants-13-00639-f006:**
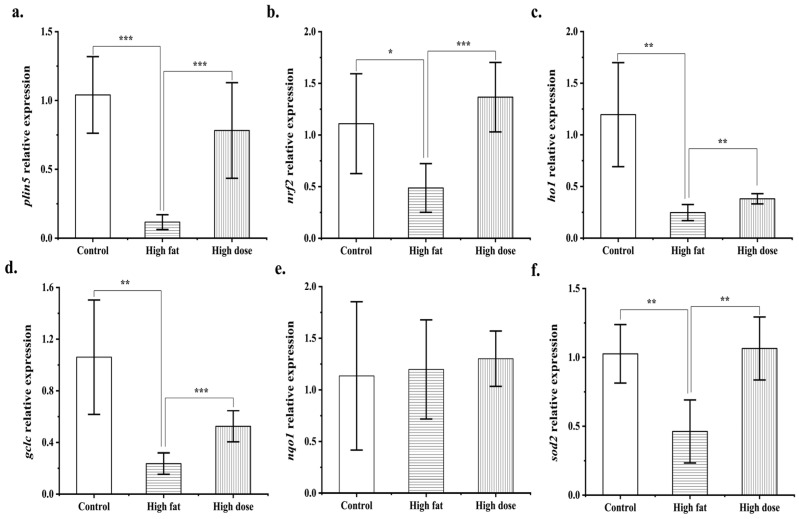
The expression level of antioxidant genes (n = 6): *plin5* (**a**); *nrf2* (**b**); *ho1* (**c**); *gclc* (**d**); *nqo1* (**e**); *sod2* (**f**). “*”, “**”, and “***” representing *p* < 0.05, *p* < 0.01, and *p* < 0.001.

**Figure 7 antioxidants-13-00639-f007:**
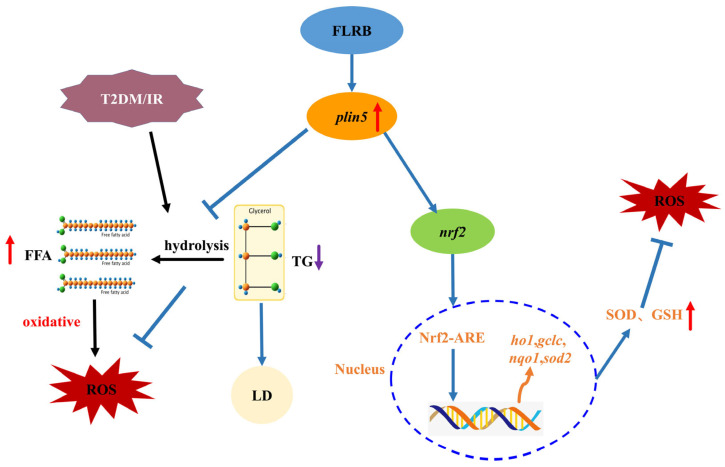
Mechanisms of antioxidant action of fermentation products in vivo, red arrows are up, purple arrows are down.

**Table 1 antioxidants-13-00639-t001:** Liquid chromatography–mass spectrometry (UHPLC-MS) analysis of fermented rice bran product (FLRB).

Number	Retention Time (min)	Molecular Weight (mass)	Component
1	4.395	148.0519	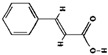 Cinnamic acid (C_9_H_8_O_2_)
2	11.028	396.1564	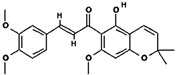 Glychalcone (C_23_H_24_O_6_)
3	11.563	340.1302	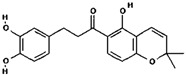 Chalcone (C_20_H_20_O_5_)
4	6.255	424.17	Pro His Gly Asp
5	6.605	507.2448	Ala Asp Phe Arg
6	6.629	561.2473	Cys Phe His Arg
7	7.516	434.2191	Ala Thr Ile Met
8	7.522	353.1408	Met Phe Gly
9	7.571	442.2062	Thr Pro Pro Glu
10	7.578	524.2575	Tyr Thr Asn Lys
11	7.592	567.2904	Phe His His Lys
12	8.202	396.2007	Ser Tyr Lys
13	8.245	439.2427	Ile Asn Pro Pro

**Table 2 antioxidants-13-00639-t002:** Differential Gene Ontology (GO) analysis classification statistical table.

GO ID	Term Type	Description	Number of Genes
GO:0009987	Biological process	cellular process	333
GO:0065007		biological regulation	302
GO:0050896		response to stimulus	209
GO:0008152		metabolic process	202
GO:0044464	Cellular component	cell part	386
GO:0043226		organelle	273
GO:0016020		membrane	236
GO:0044422		organelle part	207
GO:0044425		membrane part	206
GO:0032991		protein-containing complex	115
GO:0005488	Molecular function	binding	344
GO:0003824		catalytic activity	156

## Data Availability

The raw data supporting the conclusions of this article will be made available by the authors on request.

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
