# Peer review of "Antioxidative Potential and Ameliorative Effects of Rice Bran Fermented with Lactobacillus against High-Fat Diet-Induced Oxidative Stress in Mice"

_antioxidants, 2024, doi:10.3390/antiox13060639_

Round 1

Reviewer 1 Report

The authors presented the antioxidative potential and ameliorative effects of rice bran ferment with Lactobacillus against high-fat diet induced oxidative stress in mice. However, there are many deficiencies in the manuscript. Also, the content of the manuscript is insufficient and incomplete. The manuscript does not meet the standards of the journal, Antioxidants. Therefore, this manuscript is not recommended for publication.

1.    The text in this manuscript is difficult to understand. The authors should have the English of this manuscript checked by a native speaker.

2.    Lines 1 and 22; The title of this manuscript and the content of the abstract do not match. The authors should revise the title to accurately describe the content of the abstract.

3.    Line 60; Why didn't the authors investigate the effects of rice bran itself? Why did the authors investigate the effects of a product made by fermenting rice bran with lactic acid bacteria? If the purpose is to effectively utilize rice bran, the usefulness of rice bran itself should be verified. At present, the purpose of this study is insufficient.

4.    Line 241; Identification of constituents in FLRB by HPLC is insufficient. Have the authors identified the constituents of all the peaks appearing in the chromatogram shown in Figure S2? In this study, identification of the constituents contributing to the activity in fermented rice bran is extremely important. How did the authors identify the 13 compounds shown in Table 1? Are each constituent directly compared to standard products? The authors should isolate each constituent from fermented rice bran using chromatographic techniques and then identify them using spectral methods such as NMR. The authors should discuss biological activity using active constituents isolated from fermented rice bran. And it should be compared with the biological activity of fermented rice bran extract. The authors should mention the above points.

5.    Figure S2; The authors should add an explanation of the two chromatograms.

6.    Table S1; It should be expressed as a "solvent" rather than a "solution."

7.    Line 430; The authors also include a discussion in their explanation of the experimental results. In fact, what is included in the “Discussion” section is a “summary of this study” rather than a discussion of the study. The authors should completely revise the manuscript on the above points.

Author Response

Comments 1: The text in this manuscript is difficult to understand. The authors should have the English of this manuscript checked by a native speaker.

Response 1: Thank you for pointing this out. We tried our best to improve the English usage in the manuscript. In addition, the manuscript has been sent to AJE for language editing. We are confident that the language quality within the manuscript has been significantly enhanced.

Comments 2: Lines 1 and 22; The title of this manuscript and the content of the abstract do not match. The authors should revise the title to accurately describe the content of the abstract.

Response 2: Thank you for pointing this out. We change the title to “Antioxidant function of rice bran fermented by Lactobacillus Fermentum Mf423 and its protective mechanism against hepatic oxidative damage in high-fat diet-induced mice” in the revised draft.

Comments 3: Line 60; Why didn't the authors investigate the effects of rice bran itself? Why did the authors investigate the effects of a product made by fermenting rice bran with lactic acid bacteria? If the purpose is to effectively utilize rice bran, the usefulness of rice bran itself should be verified. At present, the purpose of this study is insufficient.

Response 3: Thank you for pointing this out. We have added an explanation in the revised manuscript in line 49-56 and 60-69.

Line 49-56, 60-69:“Rice bran, replete with proteins, dietary fiber, and a spectrum of nutrients, is an essential raw material for the development of functional foods. Despite its nutritional wealth, its utilization rate in processing is strikingly low, with a significant portion being used as poultry feed or discarded, thereby undervaluing its potential [5]. As a copious by-product of rice milling, rice bran is often underutilized, lacking in systematic research for value-added applications.  Lactic acid bacteria (LAB), renowned for their probiotic benefits, are increasingly being recognized for their role in improving food safety and promoting health, and are commonly used in food processing Fermentation with lactic acid bacteria not only promotes the release of active compounds in food resource, but also converts certain molecules into more easily absorbed metabolites, which undoubtedly improves the bioavailability of rice bran[7, 8]. Multiple studies have shown that the fermentation products of lactic acid bacteria can improve the gut microbiota, reduce cholesterol and blood sugar levels, and alleviate hypertension [9]. In addition, fermentation by lactic acid bacteria can also significantly enhance the antioxidant capacity of the fermented products. Our previous study also found that the antioxidant property,hypolipidemic and hypoglycemic activities of the products of rice bran were significantly elevated after fermentation with lactic acid bacteria[10]. Consequently, LAB fermentation presents a promising strategy for the enhanced utilization and value addition of rice bran.”

References:

[5]: Park, H. Y., K. W. Lee, and H. D. Choi. Rice Bran Constituents: Immunomodulatory and Therapeutic Activities. Food Funct, 2017, 8(3): 935-43.

[6]: Raveschot, C., F. Coutte, M. Fremont, M. Vaeremans, J. Dugersuren, S. Demberel, D. Drider, P. Dhulster, C. Flahaut, and B. Cudennec. Probiotic Lactobacillus Strains from Mongolia Improve Calcium Transport and Uptake by Intestinal Cells in Vitro. Food Res Int, 2020, 133: 109201.

[7]: Andriani, R., T. Subroto, S. Ishmayana, and D. Kurnia. Enhancement Methods of Antioxidant Capacity in Rice Bran: A Review. Foods, 2022, 11(19).

[8]: Yi, C. P., L. Xu, C. Luo, H. L. He, X. J. Ai, and H. Zhu. In Vitro Digestion, Fecal Fermentation, and Gut Bacteria Regulation of Brown Rice Gel Prepared from Rice Slurry Backfilled with Rice Bran. Food Hydrocolloids, 2022, 133.

[9]: Osei Sekyere, J., N. E. Maningi, and P. B. Fourie. Mycobacterium Tuberculosis, Antimicrobials, Immunity, and Lung-Gut Microbiota Crosstalk: Current Updates and Emerging Advances. Ann N Y Acad Sci, 2020, 1467(1): 21-47.

[10]: Ai, X., C. Wu, T. Yin, O. Zhur, C. Liu, X. Yan, C. Yi, D. Liu, L. Xiao, W. Li, B. Xie, and H. He. Antidiabetic Function of Lactobacillus Fermentum Mf423-Fermented Rice Bran and Its Effect on Gut Microbiota Structure in Type 2 Diabetic Mice. Front Microbiol, 2021, 12: 682290.

Comments 4:  Line 241; Identification of constituents in FLRB by HPLC is insufficient. Have the authors identified the constituents of all the peaks appearing in the chromatogram shown in Figure S2? In this study, identification of the constituents contributing to the activity in fermented rice bran is extremely important. How did the authors identify the 13 compounds shown in Table 1? Are each constituent directly compared to standard products? The authors should isolate each constituent from fermented rice bran using chromatographic techniques and then identify them using spectral methods such as NMR. The authors should discuss biological activity using active constituents isolated from fermented rice bran. And it should be compared with the biological activity of fermented rice bran extract. The authors should mention the above points.

Response 4:  Thank you for your suggestion. The objective of HPLC analysis was to develop a rapid fingerprinting method for Lactobacillus fermented rice bran products with high antioxidant activity. Initial isolation and purification of the fermentation product by FPLC was combined with antioxidant activity assessment and HPLC analysis in order to identify the location of the main active components in the HPLC profile. A rapid assay fingerprint was established for the stable fermentation of Lactobacillus rice bran fermentation products with antioxidant activity to meet the requirements of subsequent industrial production. In this paper, the main components of the fermentation products of Lactobacillus and rice bran without Lactobacillus fermentation were comparatively analyzed by LC-MS. Significant changes were observed for 13 components listed in table 1. Considering the cost of production and the complexity of antioxidant effects, this study focused on the overall antioxidant efficacy of Lactobacillus fermented products and provided a preliminary analysis of potential antioxidant constituents they may contain.

We greatly appreciate the reviewer's suggestions and intend to conduct further isolation and purification, as well as in-depth analytical studies of specific antioxidant-active components using NMR in our forthcoming research.

Comments 5: Figure S2; The authors should add an explanation of the two chromatograms.

Response 5: Thank you for pointing this out. We have added an explanation of Figure S2 in the revised manuscript in line 281-290.

Lines 281-290:“In order to further elucidate the changes that occurred in the products of Lactobacillus rice bran before and after fermentation, we analyzed the compositional profiles of the products using UHPLC-MS to examine the changes in substance composition. It was found that the post-fermentation peaks (Fig. S2B) showed significant changes in substance composition compared to the pre-fermentation peaks (Fig. S2A) in the time range of 4-12 min. There was a significant increase in the composition of the substance in the after-fermentation product at 4.395 min, in addition to new substances detected in the fermentation product at 6.248 min, 8.249 min, 10.319 min and 11.565 min. In the unfermented rice bran, there was a significant decrease in the fractions at 14.426 min, 14.598 min, 19.600 min, indicating degradation or transformation of the fractions.”.

Comments 7: Line 430; The authors also include a discussion in their explanation of the experimental results. In fact, what is included in the “Discussion” section is a “summary of this study” rather than a discussion of the study. The authors should completely revise the manuscript on the above points.

Response 7: Thank you for pointing this out. We had changed the "Discussion" section to "Conclusions" in revised manuscript. The results were analyzed in greater detail and the findings were extensively discussed in the 'Results and Discussion' section."

“4. Conclusion

This study introduces Lactobacillus fermentum MF423, a strain that ferments rice bran into products with potent antioxidant capabilities. An HPLC fingerprinting technique was developed to monitor the antioxidant activity of these fermentation products, ensuring reliable and direct assessment. The antioxidant efficacy of the fermentation product was confirmed through cellular tests involving various human cell types and through experiments with nematodes, where it was shown to boost antioxidant enzyme activity, suggesting potential anti-aging benefits. Our previous research indicated that FLRB can effectively ameliorate the dysregulated glucose and lipid metabolism in type 2 diabetic mice, which is closely associated with an enhancement in the liver's overall antioxidant capacity [10]. To delve into the potential mechanisms by which the antioxidant properties of the fermentation product could lead to hypoglycemic and lipid-lowering effects, we performed a transcriptomic analysis on the livers of type 2 diabetic mice that received treatment with the fermentation product. The findings revealed that the fermentation product, by upregulating the expression of the Plin5 gene, facilitates the storage of triglycerides in the liver to prevent the over-oxidation of free fatty acids and the consequent generation of ROS. Additionally, the fermentation product activates the PI3K/AKT signaling pathway through Plin5, which subsequently activates the Nrf2/ARE signaling pathway, enhancing the expression of related antioxidant enzymes downstream and thus protecting the livers of high-fat mice from oxidative damage. In conclusion, the study has developed a HPLC fingerprinting technique, which provides a solid basis for the industrial-scale production of rice bran fermented by MF423. However, there is still a need for further investigation into the optimal conditions for industrial fermentation processes. Moreover, additional research is essential to clarify the precise mechanisms through which the critical antioxidant molecules present in the fermentation product can upregulate the plin5 gene and regulate the NRF2/ARE signaling pathway.

  1. Response to Comments on the Quality of English Language

Point 1: The text in this manuscript is difficult to understand. The authors should have the English of this manuscript checked by a native speaker.

Response 1: We tried our best to improve the English usage in the manuscript. In addition, the manuscript has been sent to AJE for language editing. We are confident that the language quality within the manuscript has been significantly enhanced.

Reviewer 2 Report

The issue of paper is interesting and innovative. The paper is suitable for Antioxidatns but manuscript requires some improvement. Methods are well planned and described but references are needed. There is no conclusions which should be provided. Please  see detail comments.

Introduction: informative, based on up-to-date references

Line 93: ‘of Wang.6 The group’ please remove 6

Methods: adequate for aim of the study. Some points should be improved:

- 2.5 The fermentation products were detected by HPLC.: if possible, references could be helpful for readers.

- 2.7 The protective effect of FLRB on oxidative stress in C. elegans: references are needed

2.11 Statistical analysis: provided

Results:

Figure 1: too small, please improve font for the axis description

Table 1: molecular weight change into Molecular weight

Figure 2: please improve, the figure (especially diagrams) is hard to read

Figure 5: part a and b must be improved. There is very hard to read the axis description

Discusion: insufficient, please prepare the section in more detail

Conslusions: lack, must be provided

Author Response

Comments 1: Line 93: ‘of Wang.6 The group’ please remove 6.

Response 1: Thank you for pointing this out, We have removed 6 in lines 109.

Comments 2: 2.5 The fermentation products were detected by HPLC.: if possible, references could be helpful for readers.

Response 2: Thank you for pointing this out. We have added a reference in line 123.

line 123 : “The UV absorption detection wavelengths were set at 260, 280, and 310 nm[11].”

Reference:

[11] Punia, S., K. S. Sandhu, S. Grasso, S. Singh Purewal, M. Kaur, A. Kumar Siroha, K. Kumar, V. Kumar, and M. Kumar. Aspergillus Oryzae Fermented Rice Bran: A Byproduct with Enhanced Bioactive Compounds and Antioxidant Potential. Foods, 2020, 10(1):70.

Comments 3: 2.7 The protective effect of FLRB on oxidative stress in C. elegans: references are needed.

Response 3: Thank you for your suggestion. We have added a reference in line 134.

line 134 : “The ROS fluorescence intensity in the C. elegans was observed using an inverted fluo-rescence microscope[12]. ”

Reference:

[12] Guo, X., Q. Xin, P. Wei, Y. Hua, Y. Zhang, Z. Su, G. She, and R. Yuan. Antioxidant and Anti-Aging Activities of Longan Crude and Purified Polysaccharide (Lp-a) in Nematode Caenorhabditis Elegans. Int J Biol Macromol, 2024: 131634.

Comments 4: Figure 1: too small, please improve font for the axis description.

Response 4: Thank you for pointing this out. We've increased the font size and changed the picture in revised manuscript.

Figure 1 Hydroxyl radical scavenging activity was assessed in the products of fermented rice bran by different Lactobacillus strains at various fermentation times. Lactobacillus plantarum (a); Lactobacillus fermentum (b); Lactobacillus casei (c); "#" represents the comparison with the unfermented group, "*", "**", and "***" represents p<0.05, p <0.01, and p <0.001.

Comments 5: Table 1: molecular weight change into Molecular weight.

Response 5: We apologize for our negligence. We have changed “molecular weight“ to ”Molecular weight“ in revised manuscript.

Comments 6: Figure 2: please improve, the figure (especially diagrams) is hard to read

Response 6: Thank you for pointing this out. We have added the figure notes in Line316-321.

Comments 7: Figure 5: part a and b must be improved. There is very hard to read the axis description.

Response 7: Thank you for pointing this out. We have converted the information presented in Figure 5-a into a table format (Table 2) and have also enhanced the clarity of Figure 5 in the revised version.

Comments 8: Discusion: insufficient, please prepare the section in more detail.

Response 8: Thank you for pointing this out. We have changed the "Discussion" section to "Conclusion" and incorporated the discussion in the results section.

Line 251-253:“the retention time of the component at 16.7 minutes significantly decreased, indicating the consumption of certain substances during MF423 fermentation.”

Line 263-279:“To further elucidate the changes that occurred in the products of Lactobacillus rice bran before and after fermentation, we analysed the compositional profiles of the products using UHPLC-MS to examine the changes in substance composition. The postfermentation peaks (Fig. S2B) showed significant changes in substance composition compared with the prefermentation peaks (Fig. S2A) in the time range of 4-12 min. There was a significant increase in the composition of the substance in the postfermentation product at 4.395 min, in addition to new substances detected in the fermentation product at 6.248 min, 8.249 min, 10.319 min and 11.565 min. In the unfermented rice bran, there was a significant decrease in the fractions at 14.426 min, 14.598 min, and 19.600 min, indicating degradation or transformation of the fractions. Based on the molecular formula analysis, the main compounds generated during fermentation are outlined in Table 1. Yi et al. confirmed an increase in phenolic compounds in rice bran fermented with Lactobacillus plantarum, which exhibited increased DPPH radical-scavenging ability and ferric reducing antioxidant power (FRAP)[14]. In our current study, fermenting rice bran with Lactobacillus fermentum MF423 resulted in a notable increase in cinnamic acid, which is a typical phenolic compound. It is postulated that this phenolic compound contributes to the antioxidant activity observed in fermented rice bran products [15].”

Line 368-372:“In summary, MF423 fermentation products exhibited several beneficial effects, in-cluding extending the lifespan of C. elegans, suppressing oxidative stress induced by H2O2 and methyl viologen, reducing malondialdehyde levels, inhibiting lipid peroxida-tion, and upregulating the activities of SOD and GSH-Px. These activities collectively enhanced the antioxidant capacity of C. elegans.”

Comments 9: Conslusions: lack, must be provided.

Response 9: Thank you for pointing this out. We have changed the "Discussion" section to "Conclusion" and rewritten this section in the revised manuscript.

“4. Conclusion

This study investigated Lactobacillus fermentum MF423, a strain that ferments rice bran into products with potent antioxidant capabilities. An HPLC fingerprinting technique was developed to monitor the antioxidant activity of these fermentation products, ensuring reliable and direct assessment. The antioxidant efficacy of the fermentation product was confirmed through cellular tests involving various human cell types and through experiments with nematodes, where it was shown to boost antioxidant enzyme activity, suggesting potential anti-ageing benefits. Our previous research indicated that FLRB can effectively ameliorate dysregulated glucose and lipid metabolism in type 2 diabetic mice, which is closely associated with an increase in the overall antioxidant capacity of the liver [10]. To determine the potential mechanisms by which the antioxidant properties of the fermentation product could lead to hypoglycaemic and lipid-lowering effects, we performed a transcriptomic analysis on the livers of type 2 diabetic mice treated with the fermentation product. The findings revealed that the fermentation product, by upregulating the expression of the Plin5 gene, facilitated the storage of triglycerides in the liver to prevent the overoxidation of free fatty acids and the consequent generation of ROS. Additionally, the fermentation product activated the PI3K/AKT signalling pathway through Plin5, which subsequently activated the Nrf2/ARE signalling pathway, enhancing the expression of related antioxidant enzymes downstream and thus protecting the livers of high-fat diet-fed mice from oxidative damage. In conclusion, this study developed an HPLC fingerprinting technique that provides a solid basis for the industrial-scale production of rice bran fermented by strain MF423. However, further investigations into the optimal conditions for industrial fermentation processes are needed. Moreover, additional research is essential to clarify the precise mechanisms through which the critical antioxidant molecules in the fermentation product can upregulate the Plin5 gene and regulate the NRF2/ARE signalling pathway.

Round 2

Reviewer 1 Report

This revised manuscript has been modified according to the reviewer’s comments. It is acceptable for publication.

This revised manuscript has been modified according to the reviewer’s comments. It is acceptable for publication.

Reviewer 2 Report

The paper has been improved in accordance with my suggestions.

The paper has been improved in accordance with my suggestions.